# Conjoined Twins in Guinea Pigs: A Case Report

**DOI:** 10.3390/ani12151904

**Published:** 2022-07-26

**Authors:** Petr Tejml, Vojtěch Navrátil, Luboš Zábranský, Miloslav Šoch

**Affiliations:** Department of Animal Sciences, Faculty of Agriculture and Technology, University of South Bohemia, 370 05 Ceske Budejovice, Czech Republic; boohusek@seznam.cz (V.N.); zabransky@fzt.jcu.cz (L.Z.); soch@fzt.jcu.cz (M.Š.)

**Keywords:** guinea pig, *Cavia aperea f. porcellus*, conjoined twins, malformation

## Abstract

**Simple Summary:**

Cases of newborn connected twins, so-called Siamese twins, are well known in humans, laboratory mammals, domestic and wild birds, reptiles, amphibians and fish. This report describes a unique case of newborn conjoined twins in guinea pigs (*Cavia aperea f. porcellus*). Connected twins in guinea pigs have been very rare, and there are only a few previous reported cases. This report is only the fourth described case of the birth of conjoined guinea pigs. They were born in breeding in the Czech Republic in 2020. The conjoined twins were males connected in the upper body by a cephalopagus-type connection, which has occurred in all three previous reports. The skeletons of both the bodies were connected by a broad shared skull and sternum. Both of the fused bodies had their own hearts and livers, but the lungs were mutual. The issue of this anomaly has not yet been clearly explained.

**Abstract:**

The report describes a unique case of newborn conjoined twins in guinea pigs (*Cavia aperea f. porcellus*). Connected twins in guinea pigs have been very rare, and there are only a few previous reported cases. The newborns were stillborn in breeding in the Czech Republic in 2020 as two bodies conjoined into one. The external and internal structure of the body and the type of connection of both of the bodies are described. The weights of selected internal organs of the twins are compared with those of normally developed newborn guinea pigs.

## 1. Introduction

Cases of newborn connected twins, so-called Siamese twins, are well known in humans, laboratory mammals, domestic and wild birds, reptiles, amphibians and fish.

Numerous classifications of connected twins are based on anatomy, the place of connection, the level of symmetry of the twins and embryological development [1]. Conjoined twins are monozygotic and thus always have the same sex.

The formation of connected twins was described by two main theories [2]. The first theory assumes that, in the early stages of cell division, one fertilized egg splits into two embryos. However, the individual embryos do not split off completely as in the normal development of identical twins, but some parts of the bodies of both siblings remain connected. Conjoined twins develop when the embryo separates only partially and forms two bodies, but conjoined twins are a single organism with multiple morphological duplications. Although the two fetuses develop, they remain physically connected, most often on the chest, abdomen or pelvis. The second theory states that, at first, two embryos are completely divided, but then the active stem cells find each other and connect to form a new, differently sized and differently located adhesion [3].

The basic classification of conjoined twins is determined by the location of the anatomical connection on the body. The first discrimination in conjoined twins is the fact that some are symmetrical and others are not [4]. Most conjoined twins are symmetrical, but their internal organs are often not identical and symmetrical to each other [5]. All asymmetrical forms of conjoined twins are named heteropagus. Classically, nondorsally conjoined twins can be divided into ventral, lateral and caudal conjunction types. Ventrally conjoined twins are joined at the periumbilical regions, and, with increasing degrees of union, the thorax, neck, face and/or head can be additionally involved [6]. These are cases where one body is developed differently than the other [7]. Currently, two hypotheses of the formation of such twins are used to explain—namely, fusion and fission [1]. This fusion theory was espoused by Spencer [8] and is now a widely accepted theory.

Abnormalities in the anatomy of connected twins occur during prenatal development. Genetic and environmental factors are considered to be the main causes of the development of conjoined twins. The overall etiology of the formation of conjoined twins is still unclear [9,10].

Guinea pigs (*Cavia aperea f. porcellus*) are small mammals belonging to the order of rodents. They are native to South America, where they were domesticated mainly for consumption purposes, and they still live there in the wild. Recently, guinea pigs have been kept as laboratory and pet animals almost worldwide.

The gestation period of a guinea pig is in the range of 63–70 days. The number of neonates in a litter varies between 1 and 6 young. The young are weaned until about the fifth week of their age [11].

In spite of the high number of reared guinea pigs, it is noteworthy that records of congenital body malformations occur very rarely. One of the first cases was reported by Kaplun et al. [9]. The two bodies were joined at the site of the first cervical vertebra; they had one head, and each of them had its limbs. This deformity was the first of 12,000 newborns since 1962.

Hong et al. [10] described two cases of conjoined twins in guinea pigs at a research center in the USA. The young were dead immediately after the birth. The first ones were connected females with one head and two eyes. Another case showed male conjoined twins with a connected head and a connected front of the chest. Only these two cases occurred within a total of 288,000 newborn guinea pigs. Due to the large number of animals produced, comparisons show that genetic factors play only a marginal role in the occurrence of congenital anomalies. The case of born conjoined twins recorded in England confirms the uniqueness of such a malformation. Only this sole case of conjoined twins occurred among 4369 newborns [12].

Only a few cases of conjoined twins in guinea pigs were noticed in the Web of Science until now. A further case that appeared in breeding in 2020 is therefore described.

## 2. Materials and Methods

The basis for this report was conjoined twins born in the breeding of exhibition guinea pigs in the Czech Republic. The breed is registered under the Czech Breed Association (registration number 18041). The young pertained to a black, short-haired, smooth breed of guinea pigs. The breed has a symmetrically short and dense coat, without exceeding longer hairs. It is smooth, shiny and fits well to the body, and it does not exceed a length of 2.5 cm. The genetic line of guinea pigs is originally from England from purebred breeding. In the Czech breeding, this line is bred for five years, with the absence of inbreeding. Each guinea pig in the farm is microchipped and has a three-generation pedigree. This guarantees clear identification and provenance. In breeding, guinea pigs are kept in a special indoor room, adapted to the most suitable hygienic and microclimatic conditions. The basic space for the two kept animals has dimensions of 80 × 40 cm and is solved in the form of low plastic boxes inserted into a specially adapted metal structure with mesh. Litter consists of wood shavings.

Feed ration is based on hay and a complete feed mixture for guinea pigs with vitamin C, supplemented with selected vegetables. The guinea pigs are fed once a day. Drinking is provided from separate bottle drinkers with a volume of 500 mL, with permanent access to clean water. The temperature in the housing areas is around 20 °C, and the humidity reaches 60%.

The delivery comprising the conjoined twins began to proceed normally. The female gave delivery to three live and healthy young, all females, at intervals of 5 to 10 min. All three live young were females with birth weights of 72 g, 80 g and 82 g. The last cub followed, but it was just conjoined twins. The delivery became very complicated and lasted for 12 h. The female was administered oxytocin at a concentration of 5 IU/mL and at a dose of 0.5 mL/kg body weight in three doses of 20 min intervals, and, finally, the dead conjoined male twins were withdrawn with professional assistance. The female was subsequently excluded from the breeding. The case of the conjoined twins was quite unique in this breed out of the total number of 3078 newborns prior to the described case and 323 further neonates born until 1 February 2022.

Detailed X-ray images of the conjoined twins in several positions were taken on an X-ray apparatus Chirax 70/3 (manufacturer: Chirana, Prague, Czech Republic) with automatic digital X-ray system-FireCR Flash (manufacturer: 3D Imaging & Simulations Corp., Daejeon, S. Korea) in order to properly capture the structure of the skeleton of an anomalous individual. The newborn was weighed, and then an autopsy was carried out, focusing on the internal organs and form of the twin connection. The organs were weighed using the digital scale EMB 200-2 (manufacturer: Kern, Balingen, Germany), with a precision of 0.01 g.

## 3. Results

The conjoined young were males. Both bodies of the studied twins had their own hearts and livers. The digestive and excretory tracts were fully developed and separated in both of the bodies. Both of the bodies were connected cranially by a mutual skull (Figure 1). The skull was deformed and abnormally wide. From the shared skull, each body already had its own separate spine. The forelegs and hind legs were normally developed and completely separated in both the bodies. The fingers of both limbs were also standardly developed. The size of both the individual skeletons was relatively optimal; the newborns were not extremely long or overgrown. The joint system of the whole body did not show any obvious deviations. This is evident in the X-ray images (Figure 2 and Figure 3). The data for our conjoined twins are given in (Table 1).

## 4. Discussion

So far, only a few reports of conjoined twins in guinea pigs are available. Therefore, such malformed twins can be termed as an exceptional anomaly.

Both of the conjoined young were males. Only Hong et al. [10] reported a case of conjoined males in guinea pigs. In other described cases [9,12], the malformed twins were born as females.

In the previous report [10], the females had a subsequent gestation after the delivery of the conjoined twins and delivered only normal newborns. The second case [12] states that males and females whose newborns were the conjoined twins were previously parents of several healthy and successful litters. These statements are largely consistent with our report. The female had one previous litter of three live and healthy young with birth weights of 85 g, 90 g and 95 g. Two of them were males, and the third one was a female. The first litter of the female was at the age of 9 months, and the second one with twins was at the age of 16 months. All of them had the same father.

There was only one sternum in the conjoined twins, and separate ribs continued from the sternum for each body separately. According to Spencer [3], it can clearly be determined that these are conjoined twins of the cephalopagus type, characterized by a common large head and a part of or the entire chest. This type of conjoined twins is nonviable [13]. In each of the three mentioned reports [3,10], the adhesion of the anterior body was always present, while the posterior part was duplicated and developed independently. It can thus be said that the connection type cephalopagus has prevailed among conjoined twins in guinea pigs. Only López et al. [14] described another type of physical malformation, in which two bodies have not been joined in the form of cephalopagus. However, in this case, it is a typical connection of two guinea pig newborns.

Elward and Ruelokke [15] state that the birth weight of guinea pigs is between 50 and 130 g. Nevertheless, newborns weighing less than 60 g rarely survive. The conjoined twins weighed 136.0 g. This weight is closer to the upper limit of the weight range of a single newborn young. This naturally explains the reason for the complicated delivery, when the already very unusually deformed body had a high weight as well, although it was average in size. However, if we take into account that the conjoined twins are in fact two bodies, the weight of each body would rank them among normal and lighter newborns.

Both bodies of the studied twins had their own hearts and livers. This was the same in the case recorded in England [12]. In contrast, Hong et al. [10] reported one mutual heart and one connected liver. Unfortunately, the published cases of the conjoined twins did not provide any specific weight parameters of individual organs that could be used for a comparison.

The digestive and excretory tracts were fully developed and separated in both of the bodies. This is related to the cephalopagus type of body connection, where the deformations manifest themselves mainly in the upper parts of the bodies, but the lower part of the body is usually independent and fully developed.

In the case hitherto described [10], the total deformation of the facial parts—eyes, nose and mouth—is reported. For cephalopagus type connections, the occurrence of a single central eye on the common facial part was depicted [12]. Our case of the conjoined twins is not an exception. Both facial parts were generally merged into one mutual unit. The connected face was very deformed, with almost unrecognizable individual organs. This state can be explained as a consequence of the complicated delivery.

Congenital physical malformations caused by some plant toxins, mercury poisoning and teratogenic substances have been observed in domestic animals. External causes of these abnormalities also include radiation, infections, hyperthermia, hypothermia, toxic metals or adverse maternal conditions such as obesity, diabetes, iodine deficiency, etc. [14].

All guinea pigs in the given breed received the same degree and quality of care, including feed. The mentioned external factors thus appear to be unlikely to cause the deformations. The animals received a very high-quality and balanced feeding ration. A further probable cause mentioned in the literature could be obesity [14]. Older guinea pigs can often be prone to it. The female, however, was not obese.

The second real cause of malformations can be found in our case in the exposure of guinea pigs to high outdoor temperatures. This was reported by López et al. [14].

## 5. Conclusions

The occurrence of newborn conjoined twins is always very interesting and unique. Due to the high number of reared guinea pigs, it is remarkable that congenital malformations occur very rarely.

This report is only the fourth described case of the birth of conjoined guinea pigs. They were born in breeding in the Czech Republic in 2020. The conjoined twins were males connected in the upper body by a cephalopagus-type connection, which has occurred in all three previous reports. The skeletons of both of the bodies were connected by a broad shared skull and sternum. Both of the fused bodies had their own hearts and livers, but the lungs were mutual. The issue of this anomaly has not yet been clearly explained. Due to the large number of newborns, it seems likely that genetic factors are negligible in the occurrence of congenital anomalies. However, breeding conditions—mainly, extreme ambient temperature, housing or nutrition—can also be hypothetically included among the causes.

## Figures and Tables

**Figure 1 animals-12-01904-f001:**
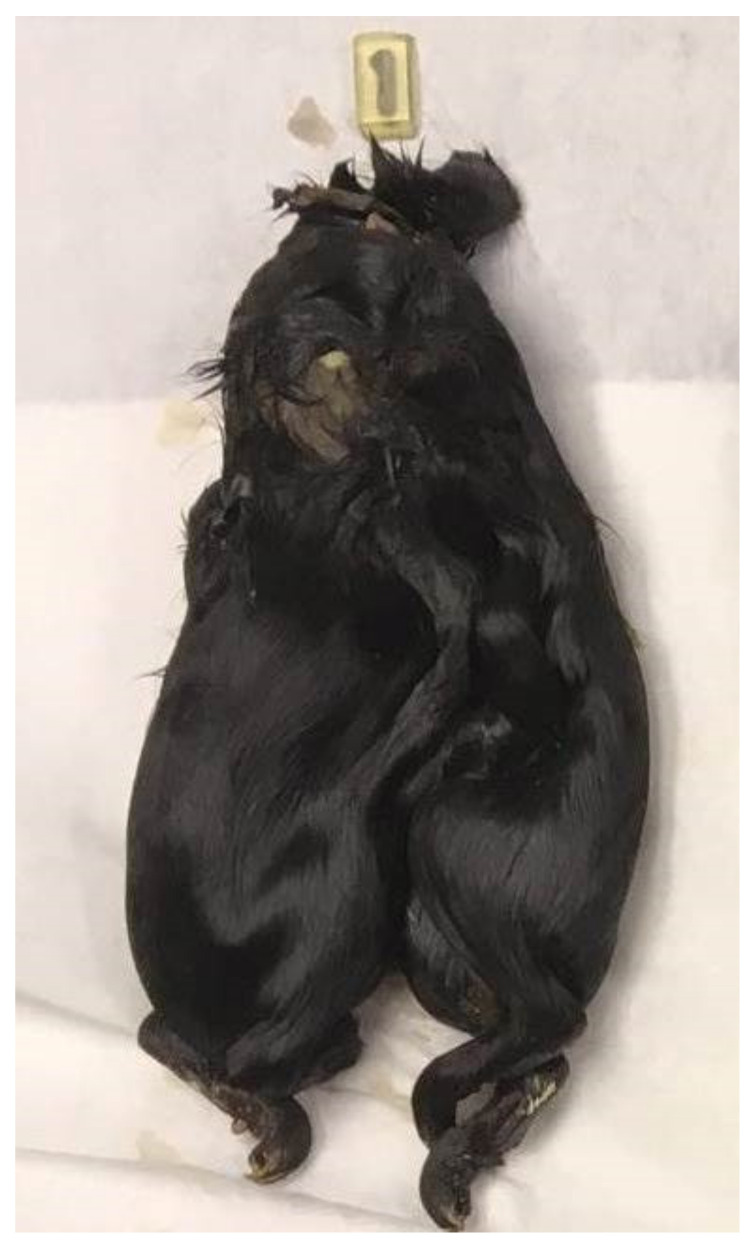
The conjoined young.

**Figure 2 animals-12-01904-f002:**
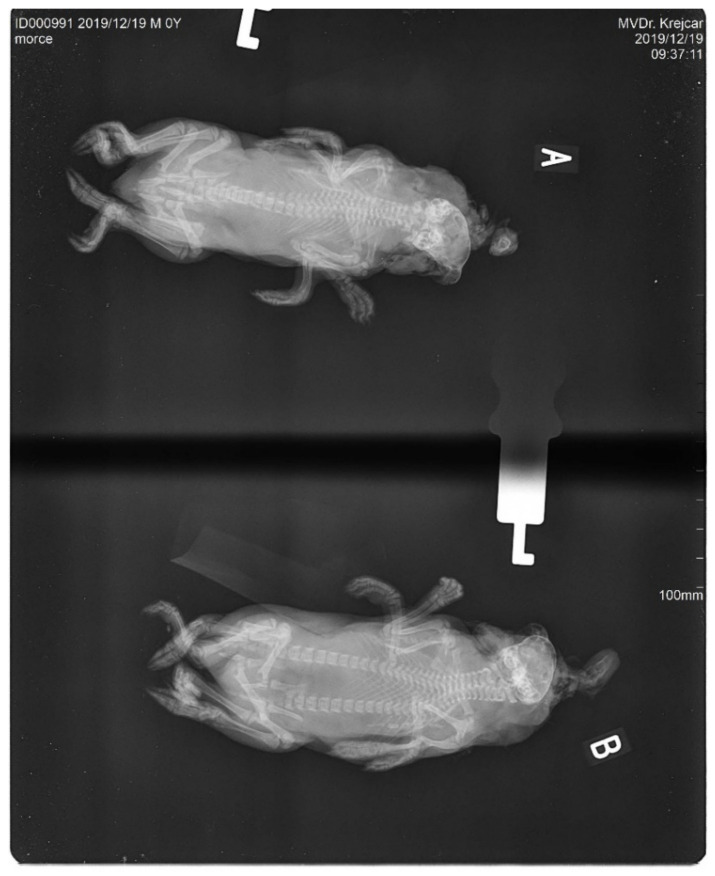
The X-ray images—AP projection (A -anterior, B- posterior).

**Figure 3 animals-12-01904-f003:**
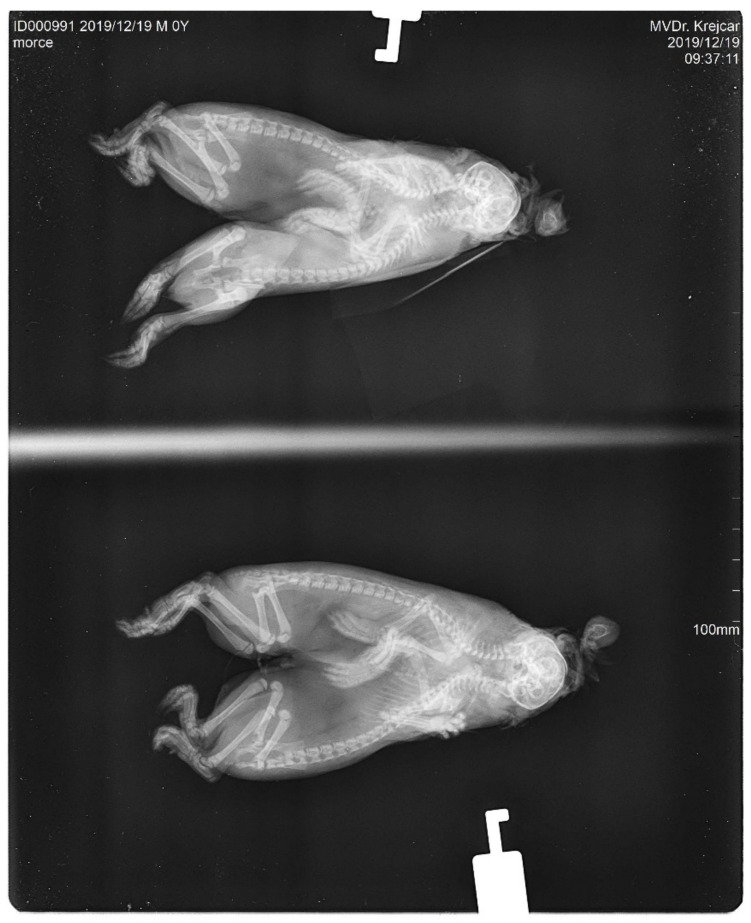
The X-ray images—side projection (left-right).

**Table 1 animals-12-01904-t001:** Organ weight (g) of the conjoined twins and usual values of healthy newborn guinea pigs. Usual values for the described breed were obtained from ten newborns born dead. The bodies were dissected, and the organs were separated and weighted.

Organs	Conjoined Twins			Usual Values [11]	Standard Deviation
Newborn	136.0			108.5	32.03
Heart	0.25	0.34	Σ 0.59	0.68	0.24
Liver	1.97	3.39	Σ 5.36	5.62	2.77
Lungs (mutual)	0.37			1.26	0.58

## Data Availability

The data that support the findings of this study are available from the corresponding author upon reasonable request.

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
