# Peer review of "Conjoined Twins in Guinea Pigs: A Case Report"

_animals, 2022, doi:10.3390/ani12151904_

Round 1

Reviewer 1 Report

The authors describe the case of conjoined guinea-pig twins. Under general aspects, this is worth to become published. However, the reviewer recommends some improvements:

- The description of the genetics of this guinea-pig line is very poor

- If the reviewer understands the authors right, the conjoined twins were delivered together with three healthy siblings. Please add more details

- Please add more details about the mother and previous pregnancies

- The calculations in Table 1 are a bit confusing. The values of helthy animals should be verified with published data. The corresponding values of the healthy siblings of the conjoined twins should be added.

- The Figure legends should give a lot of information. E.g. "The X-ray images" are not very helpful.

- Please explain better where the animals were housed. "in the breeding in the Czech Republic" is not very transparent.

- Case reports are according to the guidelines of the journal limited to 2500 words, the rough estimate of the reviewer is at least 3200 words, i.e. 20% more.

- Please avoid redundancies, the introduction might be condensed

Author Response

Dear reviewer,

thank you very much for your email with information concerning our manuskript „Conjoined Twins in Guinea Pigs: A Case Report“.

We read the coments of reviewers very carefuly. The fallowing part of this letter includes the reactions of the authors to individual requests and the notes of the reviewers.

We houpe that all the comments were acknowledged and corrected accordinagly. Some parts of the text, mentioned in the peer reviwev were changed, delected or added according to requirements of reviewers.

Answers to reviewers questions

- The description of the genetics of this guinea-pig line is very poor

The genetic line of guinea pigs is originally from England from purebred breeding. In the Czech breeding, this line is bred for five years with the absence of inbreeding. Each guinea pig in the farm is microchipped and has a 3 generation pedigree. This guarantees clear identification and provenance.

Line 85

- If the reviewer understands the authors right, the conjoined twins were delivered together with three healthy siblings. Please add more details

Yes. There were 3 other youngs in the litter that were born as first and alive. A double young followed. All 3 live youngs were females with birth weights of 72 g, 80 g and 82 g. The litter was reared successfully by their mother.

Line 101

- Please add more details about the mother and previous pregnancies

The female had one previous litter of three live and healthy youngs with birth weight 85 g, 90g and 95g. Two of them were males and the third one was a female.  The first litter of the female was at her age of 9 months and the second one with twins at the age of 16 months. All of them had the same father.

Line 157

- The calculations in Table 1 are a bit confusing. The values of healthy animals should be verified with published data. The corresponding values of the healthy siblings of the conjoined twins should be added.

Table updated, line 142

- The Figure legends should give a lot of information. E.g. "The X-ray images" are not very helpful.

Figure 2 AP projection (anterior - posterior), line 133

Figure 3 Side projection (left - right), line 137

- Please explain better where the animals were housed. "in the breeding in the Czech Republic" is not very transparent.

The breed is registered under Czech Breed Association, registration number 18041.

Line 82

- Case reports are according to the guidelines of the journal limited to 2500 words, the rough estimate of the reviewer is at least 3200 words, i.e. 20% more.

- Please avoid redundancies, the introduction might be condensed

Thank you for your suggestion. However, due to the expertise of the issue, the information is necessary. The other opponent, on the other hand, requested an extension.

Thank you for Your cooperation and suggestions. We think that all changes helped to enhance scientific level of our manuskript.

Than you very much for Your time and cooperation.

Sincerely

Petr Tejml

Reviewer 2 Report

Dear authors, 

This is a very interesting article concerning a conjoined guinea pig. Please find some minor comments below:

page 1. line 19. born dead = stillborn 

page 1. line 32-40 I do miss some references. Please check Boer et al., 2019 "Two is a crowd" for some general background information about conjoined twins in general 

page 1 line 36. two individuals implies some sort of separate entities. It is better to describe conjoined twins as a single organism with multiple morphological duplications 

page 2. Line 45. the fission and fusion theory are both somewhat outdated, although often used. If used these theories are also thought to produce symmetric twins. The only difference in heteropagi vs. symmetric twins is the fact that one became (hemodynamically) unstable and lacked further outgrowth. they are always located at the same areas as symmetrical conjoined twins. Please consider further documentation in the literature, some additional references should be included on this part.

page 2 line 47.  Characterizing the coalescence area of conjoined twins to elucidate congenital disorders in singletons' for some general information about secondary malformations in twins. Now, i miss a reference and this statement is somewhat general. 

page 7 line 139: This case concerns a cephalothoracoileopagus tetrabrachius tetrapus with extreme lateral deviations at its cranial region creating the illusion of a single and mutual ventrally located facial region. Please consider to include additional literature  for background information about this type. 

page 7 line 146, what is meant with dimorphism? 

Overall this is an interesting report, however, I do miss some recent etiopathogenetic thoughts about how these rare entities potentially arose. It would be very nice to include some of these thoughts.

Author Response

Dear reviewer,

thank you very much for your email with information concerning our manuskript „Conjoined Twins in Guinea Pigs: A Case Report“.

We read the coments of reviewers very carefuly. The fallowing part of this letter includes the reactions of the authors to individual requests and the notes of the reviewers.

We houpe that all the comments were acknowledged and corrected accordinagly. Some parts of the text, mentioned in the peer reviwev were changed, delected or added according to requirements of reviewers.

Answers to reviewers questions

page 1. line 19. born dead = stillborn

Reworked.

line 20

page 1. line 32-40 I do miss some references. Please check Boer et al., 2019 "Two is a crowd" for some general background information about conjoined twins in general

The information added briefly, as the other opponent requested shortening the text.

line 43, 46

page 1 line 36. two individuals implies some sort of separate entities. It is better to describe conjoined twins as a single organism with multiple morphological duplications

The information added briefly, as the other opponent requested shortening the text.

line 36

page 2. Line 45. the fission and fusion theory are both somewhat outdated, although often used. If used these theories are also thought to produce symmetric twins. The only difference in heteropagi vs. symmetric twins is the fact that one became (hemodynamically) unstable and lacked further outgrowth. they are always located at the same areas as symmetrical conjoined twins. Please consider further documentation in the literature, some additional references should be included on this part.

The information added briefly, as the other opponent requested shortening the text.

Line 51, 43

page 2 line 47.  Characterizing the coalescence area of conjoined twins to elucidate congenital disorders in singletons' for some general information about secondary malformations in twins. Now, i miss a reference and this statement is somewhat general.

Reference added.

Line 56

page 7 line 139: This case concerns a cephalothoracoileopagus tetrabrachius tetrapus with extreme lateral deviations at its cranial region creating the illusion of a single and mutual ventrally located facial region. Please consider to include additional literature  for background information about this type.

The information added briefly, as the other opponent requested shortening the text.

Line 164

page 7 line 146, what is meant with dimorphism?

Bad text formulation. Thank you for the notice. Reworked.

Line 169

Overall this is an interesting report, however, I do miss some recent etiopathogenetic thoughts about how these rare entities potentially arose. It would be very nice to include some of these thoughts.

Line 193

Thank you for Your cooperation and suggestions. We think that all changes helped to enhance scientific level of our manuskript.

Than you very much for Your time and cooperation.

Sincerely

Petr Tejml

Round 2

Reviewer 1 Report

The reviewer feels that the manuscript qualifies for publication now. However, according to the guidelines for case reports of the journal the size is a bit too large.